# Molecular Mechanisms at the Basis of Pharmaceutical Grade *Triticum vulgare* Extract Efficacy in Prompting Keratinocytes Healing

**DOI:** 10.3390/molecules25030431

**Published:** 2020-01-21

**Authors:** Antonella D’Agostino, Anna Virginia Adriana Pirozzi, Rosario Finamore, Fabrizia Grieco, Massimiliano Minale, Chiara Schiraldi

**Affiliations:** 1Department of Experimental Medicine, Section of Biotechnology, Medical Histology and Molecular Biology, University of Campania “Luigi Vanvitelli”, 80138 Naples, Italy; antonella.dagostino@unicampania.it (A.D.); annavirginiaadriana.pirozzi@unicampania.it (A.V.A.P.); rosario.finamore@unicampania.it (R.F.); 2Farmaceutici Damor, 80145 Naples, Italy; fabri.gri1224@hotmail.it (F.G.); massimiliano.minale@farmadamor.it (M.M.)

**Keywords:** *Triticum vulgare* aqueous extract, wound repair, time lapse video microscopy, matrix remodeling

## Abstract

**Background**: It has been shown that many plant- or microbial-derived oligos and polysaccharides may prompt tissue repair. Among the different extracts that have been studied, the aqueous one of *Triticum vulgare* (TVE) that was obtained from a whole germinated plant has been proven to have different biological properties that are useful in the process of wound healing. Nevertheless, with the long tradition of its use in pharmaceutical cream and ointments, especially in Italy, a new protocol was recently proposed (and patented) to improve the extraction process. **Methods**: In a simplified in vitro model, human keratinocyte monolayers were scratched and used to run time lapse experiments by using time lapse video microscopy (TLVM) to quantify reparation rate while considering a dose–response effect. Contemporarily, the molecular mechanisms that are involved in tissue repair were studied. In fact, key biomarkers that are involved in remodeling, such as MMP-2 and MMP-9, and in matrix structure assembly, such as collagen I, elastin, integrin αV and aquaporin 3, were evaluated with gene expression analyses (RT-PCR) and protein quantification in western blotting. **Results**: All TVE doses tested on the HaCat-supported cell proliferation. TVE also prompted cell migration in respect to the control, correctly modulating the timing of metalloproteases expression toward a consistent and well-assessed matrix remodeling. Furthermore, TVE treatments upregulated and positively modulated the expression of the analyzed biomarkers, thus resulting in a better remodeling of dermal tissue during healing. Conclusions: The in vitro results on the beneficial effects of TVE on tissue elasticity and regeneration may support a better understanding of the action mechanism of TVE as active principles in pharmaceutical preparation in wound treatment.

## 1. Introduction

Plant extracts have demonstrated beneficial effects in wound-healing, promoting skin repair through the improvement of bio-adhesive ability, immunomodulation, cell–cell and cell–matrix interactions, and collagen synthesis [1]. Recently, a positive effect in dermal tissue reparation was found in an *Opuntia cladode* extract; in particular, chemical characterization unraveled the presence of the mucilage (polysaccharide) as well as the presence of a fraction containing low molecular weight components that proved active in a wound repair in vitro model [2]. Among many others examples, an aqueous extract of *Triticum vulgare* (TVE) obtained from the whole germinated plant and containing mainly poly/oligosaccharidic components has different biological properties, ultimately acting as bioactive complex that directly interacts with wound repairing factors [3]. This extract has been extensively used in gels, ointments and spray to accelerate tissue repair. In fact, TVE-based products speed up healing processes both in cutaneous and not cutaneous tissues [4,5]. In particular, scientific evidence has documented that wheat sprout oil has been used in traditional Iranian medicine for dermotonic and skin beauty, face freckles, and the moisturizing and repair of the minute pores of face skin [6]. TVE is commonly used for the treatment of decubitus ulcers, venous leg ulcers, sores, burns, scarring delays, dystrophic diseases, and, more broadly, in the presence of problems that are related to re-epithelialization or tissue regeneration [4,7,8,9]. This extract is currently an active component in the brand name Fitostimoline^®^ (Farmaceutici Damor S.p.A., Naples, Italy). The active components of Fitostimoline^®^-based products prompt hastening tissue repairing processes, stimulate chemotaxis and fibroblastic maturation, and significantly increase the fibroblastic index, which are crucial points in the repairing processes [10,11]. It has been suggested that these activities are due to accelerated protein synthesis, enhanced captation, and enhanced incorporation ability of marked proline from tissues [12]. These products, whose active ingredient is obtained through a patented process, have been used in the last few years in the treatment of cutaneous lesions in which the stimulation of repairing processes (e.g., ulcerative-dystrophic damages, burns, and delay in scaring) is needed, and their efficacy in combined therapeutic approaches is well-recognized. Concerning natural products, recent literature has reported a beneficial effect of *Triticum aestivum* on dermatological diseases performed on a HaCat cellular model [13,14].

Recently, it has been reported that a specific aqueous extract of *Triticum vulgare*, named Rigenase^®^, which has shown a potent antioxidant capacity, can exert a relevant radical scavenging activity to accelerate repair in injuries like burns that are characterized by a high concentration of free radicals [15]. The specific extract of *Triticum vulgare* (TVE) is manufactured by a recently implemented extraction process (U.S. Patent No. 9,895,392) of Farmaceutici Damor, and it is used as active principle in pharmaceutical formulations for the treatment of decubitus ulcers, skin lesions and burns; it has also shown anti-inflammatory properties for reducing nitric oxide, IL-6, PGE2 and TNFα in a rat microglia cell model [16,17]. Based on all these observations reported in the literature, we aimed to assess the biochemical mechanisms at the basis of the beneficial effects of TVE, exploiting its well-assessed cell model based on scratched HaCat monolayers. Specifically, the biological effects of TVE were evaluated through a robust quantitative analysis by using time lapse video microscopy (TLVM), followed by a study of key biomarkers to unravel (or to give an insight into) the molecular mechanisms that are involved in the tissue repair at gene (MMP-2, MMP-9, collagen I and elastin) and protein (integrins, collagen I, elastin, and aqp3) levels.

## 2. Results

### 2.1. TVE Chemical and Biological Characterizations

#### 2.1.1. TVE Analytical Characterization

The hydrodynamic characterization of TVE was accomplished by size exclusion chromatographic (SEC)-TDA. The sample was analyzed at a theoretical concentration of 6 g/L to obtain a desired column load with reference to the mass of the oligosaccharide [12]. The results showed the the first peak (not reported in Table 1) accounted for 2.6% of the total carbohydrate fraction and presented an average molar mass (M_w_) of 373.8 kDa; while, as shown in Table 1, two other main peaks named peaks I and II with representativity values of 28% and 69%, respectively with an M_w_ of 6.2 kDa were detected with a total sample recovery of 82.2%.

The amount of monomeric sugars residues were analyzed by HPAE-PAD; specifically, glucose and xylose were found, respectively, at concentrations of 0.13 ± 0.02 and 0.065 ± 0.011 g/L, equal to 2.23% and 1.08% in respect to the total amount of dry matter present in the extract.

#### 2.1.2. Cell Viability Assay

TVE was able to support growth at all the dilutions tested. As shown in Figure 1, a cell viability increase was evident after 24 and 48 h. In particular, the TVE that was supplemented at a range between 5% and 3% *v*/*v* to the medium proved to have the best efficiency.

#### 2.1.3. In Vitro Keratinocytes Scratch Assay by TLVM

Wound closure was evaluated by using TLVM, and the results are reported in Figure 2. In particular, Figure 2a shows a representative picture of the monolayers during the healing process at different times (0, 6, 12, 24, and 48 h) for the control and the sample in the presence of TVE. The images show that all the treatments sped up the reparation in respect to the control. Quantitative analyses (Figure 2b) confirmed the qualitative data of the images. At an early stage, it could not be proved that a difference existed between the treatments and the control. However, after 10 h, the difference became significant, especially the samples treated with 10% and 5% *v/v* TVE, which proved superior to control. Furthermore, we observed a slightly but not significant difference between the 10% and 5% TVE, though only in the range of 10–20 h. A confluence of 80% was reached by the nineth hour of the experiment for the 5% *v/v* TVE treatment, while 11 h were necessary it, on average for the 10% treatment, and the 3% treatment was slightly slower though significantly better than the control in achieving the scratch full repair.

#### 2.1.4. Biomarker Regulations at Transcriptional Level Analyzed by Real Time PCR

MMP-2 gene expression was found to be increased after 6 h of treatment by all TVE-based treatment, and it decreased after 24 h in respect to the control, in which the expression level was upregulated at 24 h, probably due to the late repair of the scratch in the control that still needed matrix remodeling to obtain keratinocyte migration. Additionally, MMP-9 was modulated by TVE addition, but the time frame of modulation was delayed. In fact, in the control samples, MMP-9 expression was higher at 24 h than at 6 h, and the upregulation was less evident (Figure 3) during all treatments. During healing, the assembly of the key matrix protein may have delivered a functional tissue, while the production of collagens or dis-homogeneous matrix component expression may have led to fibrosis. For this reason, collagen I expression was evaluated via RT-PCR, and it was found to be increased by treatments after 24 h, a time frame that is coherent with the timing of repair, as found in the TLVM experiments (or scratch test). Elastin was expressed only at 24 h of treatment, and the beneficial effect of TVE was evident specifically at low doses (3% *v*/*v*) (Figure 4).

#### 2.1.5. Evaluation of Collagen Type I, Elastin, Aqp3 and Integrin αV as Biomarkers in Wound Healing Model

In presence of TVE 5% and 3% *v/v,* collagen I, elastin and integrin αV were also increased in respect to the control. In addition, the water channel protein Aqp3, a specific biomarker for water transport in the direction of an osmotic gradient, was upregulated, thus indicating a potential positive role in of TVE in the dermal regeneration (Figure 5 and Figure 6).

## 3. Discussion

Traditional medicine in Eastern and Western countries often use plant extracts to address specific health issues, claiming the beneficial effects of leaves, flowers and seeds. A large portion of the scientific research on novel ingredient still focuses on natural extracts, especially because the complexity and variability of the natural molecules often need deep characterization studies regarding the specific structure–function relation, as well as the assessment of a dose-dependent effect. It has been shown that many plant-derived or microbial-derived polysaccharides may play beneficial roles in tissue repair. Bio-adhesive properties may be achieved in commercial formulations with the use of additives, while active principles may be effective at low doses. The mechanism of action in a generally agreed view is involved at different levels of healing, and is a more complex process in vivo. In fact, starting from an inflammatory prompting of cell recruitment (e.g., monocytes), and there is a stimulation of metalloproteinases biosynthesis to degrade extracellular matrix proteins, thus favoring cell migration towards healing. However, there are pathologies in which chronic scars are caused by the persistent expression of such metalloproteinases. In this respect, the scientific community has agreed that not only the kind of biomarkers, the extent of protein expression, and the extent glycosaminoglycans expression but also the timing for the biosynthesis and the assembly of extracellular matrix ECM components are effective and important toward the rebuilt of a functional tissue. This has been specifically well-assessed for dermal repair. In fact, TVE has proven to improve the healing of superficial damaged tissues; specifically, a reduction in the lesion surface area has been observed, and clinical signs (perilesional erythema and bleeding) and symptoms (burning, pain, and itching) of venous leg ulcers have been shown to improve [7]. In this respect, notwithstanding the long tradition of safe and effective use of TVE in commercial formulation, we aimed at investigating the extract of *Triticum vulgare* that was obtained through a patented process (using good manufacturing practice procedures) in a well-established model of in vitro wound healing. To the best of our knowledge, the beneficial effects of this kind of extract on fibroblast have been already established; however, no specific study on keratinocytes has been available up to date. The TLVM station allowed for the robust analyses of dose–response assessments, and the TVE showed a beneficial effect in speeding up wound healing rate, specifically at 5% and 10% *v*/*v*. In the TVE-treated samples, the time needed to reach a repair of 60% was significantly reduced in the scratched area. Furthermore, the 5% and 10% treatments were significantly better than the 3% treatment to reach 80% scratch repair. In addition, we attempted to unravel the molecular features beyond this beneficial effect, and we evaluated the key biomarkers that involved in the tissue repair. Recently, Bassino and collaborators [18] found that plant extracts modulated MMPs expression by using scratched HaCaT cells in an in vitro model. In fact, that natural compounds that contained flavonoids changed the MMP-9 expression at all doses tested, though only during wound repair, and were not modulated in physiological conditions. In our study, we also investigated the modulation of MMP-2 at the gene level to deepen knowledge on the molecular determinants, thus prompting wound closure towards healthy and functional dermal tissues. Specifically, the most frequently determined matrix metalloproteinases, namely MMP-2 and MMP-9 (overexpressed in 54.5% of wounds, as reported by Tardáguila-García et al. 2019 [19]) were expressed at the transcriptional level with a specific time frame that was supportive and coherent regarding the scratch repair stage. In particular, MMP-9 modulation has been found to be less evident in respect to MMP-2, which may due to the temporal range chosen (6 or 24 h). Specific biomarkers of healing also include integrins, which are matrix transmembrane glycoproteins, that are involved in the recruitment of regulatory elements that are important for cytoskeletal reinforcement, proliferation, polarization, and migration [20]. We observed a modulation of αV integrins, and, specifically, their production increased even at low TVE concentrations (3%). At the same concentration, TVE also prompted collagen I and elastin biosynthesis in agreement with the integrins’ role as a platform for the adhesion and organization of the collagen and elastin fibers [21]. Even though fibroblasts are generally appointed as major collagen producers, few reports have evaluated the modulation of this matrix protein in an HaCaT cell model. Specifically, a superior expression of collagen I has been previously reported in contexts when the cells were treated with other compounds [22,23,24]. Additionally, Aqp3, a water channel that is required to promote glycerol permeability and water transport across cell membranes in skin, has been found to be upregulated by TVE treatment, thus indicating a positive effect on skin hydration and keratinocyte proliferation, as reported by Quin and collaborators [25]. Finally, coherently with the specific stages of the healing, key matrix biomolecules were prompted by the TVE, resulting in a faster reparation rate and thus possibly reducing the occurrence of infection contemporary permitting the assembly of an elastic and functional dermal tissue (keratinocyte layer). In vitro studies indeed have limitations, but, in this specific case, the in vivo evidence that has been reported for decades may benefit the deepening of scientific approaches and molecular mechanisms that, in turn, help in developing novel formulations for specific uses.

## 4. Materials and Methods

### 4.1. Procedure to Obtain Triticum Vulgare Extract (TVE)

*Triticum vulgare*, the binomial scientific name of a plant of the Graminaceae family, is the commonly known wheat plant. It is grown under controlled conditions in the laboratory of Farmaceutici Damor, located in Naples, Italy. The reference specimen is DF/237/2014, and it is deposited in the herbarium of the Medical Botany Chain of University of Salerno, Fisciano, Italy. The commercially available seeds were purchased from Consorzio Agrario Lombardo Veneto from Northern Italy. The batch number for the seeds used for the present paper was B00369/2017A001. TVE is an aqueous extract of *Triticum vulgare* that is obtained through a complex and patented process [16]. It was prepared according to a standard manufacturing protocol of germination under restricted conditions of temperature, light and humidity, extraction using water, sulfuric and chloride acid, and purification with a membrane filter [26].

### 4.2. Analytical Characterization of TVE

#### 4.2.1. SEC-TDA

Technical data have shown that TVE contains a dry residue of about 611 mg (% dry wt), 4% of which (*w*/*w*) is amount for the active components. The molecular distribution of oligosaccharide fraction in TVE extract was evaluated by using a high performance size exclusion chromatographic system (SEC-TDA, Viscotek, Malvern, UK) that was equipped with a triple detector array module, including a refractive index (RI), a four-bridge viscosimeter (VIS), two laser right-angle light scattering (RALS), and low-angle light scattering (LALS) detectors, as previously reported [27]. The dn∙dc−1 values used were 0.151 mL·g^−1^, which in agreement with a starch/maltodextrins-based composition. The representativeness of each peak in percentage was calculated as the percentage ratio between the single peak area divided by the sum of areas of all the peaks in each chromatogram. The amount of the carbohydrate fraction in respect to the dry weight was also evaluated. For each of the peak M_w_, M_w_/M_n_, and intrinsic viscosity were analyzed with the specific features of the software and the Mark–Hownick correlations [27].

#### 4.2.2. Monosaccharide Composition Analyses by HPAE-PAD

The TVE extract that contained the oligosaccharide mixture was ultrafiltered on a 3kDa cut-off membrane to separate the very low molecular weight fraction (e.g., glucose and xylose). The permeate containing monosaccharide was analyzed by a high pressure ion chromatography system (ICS3000, Thermo Fisher Scientific, Milan, Italy) that was equipped with a pulsed amperometric detector (reference electrode Ag–AgCl; measuring electrode Au) and anion exchange column (Carbopac PA1, Thermo Fisher Scientific, Milan, Italy). Isocratic conditions (NaOH 16mM) were used, 25 µL of sample were injected at a flow rate of 0.9 mL/min and a temperature of 25 °C.

### 4.3. Cell Culture and Treatments

HaCaT, a spontaneously transformed non-tumorigenic human keratinocytes cell line, was provided by Istituto Zooprofilattico, (Brescia, Italy), and the cells were cultured in Dulbecco’s Modified Eagle Medium (DMEM), supplemented with 10% (*v*/*v*) heat-inactivated fetal bovine serum (FBS), penicillin 100 U/mL and streptomycin 100 μg/mL (Pen-Strep). DMEM, FBS, Pen-Strep, Phosphate buffer saline PBS and trypsin were provided by Gibco Invitrogen (Milan, Italy).

The *Triticum Vulgare* extract (TVE) was received from Damor and was concentrated 3-fold in respect to the one used in the commercial formulations. Previous literature has shown that the optimal concentrations that are used in in vitro experiments are 5% and 10% of TVE [12]. For this reason, we decided to use TVE that was opportunely diluted in physiological buffer or cell medium, depending on our requirements, to obtain final concentrations of 10%, 5% and 3% *v*/*v*.

### 4.4. Cell Viability Assay

Cells were seeded at a density equal to 2 × 10^4^ cells/cm^2^. Cells were treated with TVE at 10%, 5% and 3% *v*/*v*. After 24 and 48 h of treatments, the medium was removed, and, after exhaustive washing in a physiological solution, cells were observed at inverted optical microscope. Then, cell viability was evaluated with Presto Blue assay (Cat. N. A13261, Invitrogen, GIBCO, Gaithersburg, MD, USA) according to the manufacturer’s instructions. The viable cells reduce resazurin into resofurin; this conversion was proportional to the metabolically active cells, and it was quantitatively determined by absorbance measurements at 570 nm by using a UV/Visible Spectrophotometer (Beckman Coulter, Milan, Italy).

### 4.5. In Vitro Keratinocytes Scratch and TLVM Assay

A wound healing assay was performed as previously described [24,28]. Briefly, keratinocytes were seeded in 12-well until cellular confluence. After scratching each of the 2D layers in the multi-well system, we added the following treatments: TVE 10%, 5% and 3% *v/v* (triplicates for each sample) in a DMEM 1% FBS solution. In vitro cell migration was performed by video microscopy time lapse experiments (TLVM) (OKOLAB, Pozzuoli, Italy). This technique allowed us to select and record representative images of the experiments and to follow the wound reparation for 48–72 h. Successively, qualitative and quantitative analyses of the wound closure were accomplished for the control samples (only scratch) and for the scratched monolayers that were treated with the different concentrations of TVE.

### 4.6. Metalloproteinases and Matrix Proteins Gene Expression: Real-Time PCR Analysis

Scratched keratinocyte monolayers were treated with TVE 10%, 5% and 3% *v/v* for 6 and 24 h. Total RNA was isolated from human dermal keratinocytes by using TRIzol^®^ (Invitrogen, Milan, Italy), according to the manufacturer’s procedures. Briefly, one μg of DNase-digested total RNA (DNA-free kit; Ambion-Applied Biosystems, Foster City, CA, USA) was converted to cDNA with a Reverse Transcription System Kit (Promega, Milan, Italy). PCR was then performed by using iQ™ SYBR^®^ Green Supermix (Bio-Rad Laboratories Srl, Milan, Italy) with the appropriate primers to quantify the expression levels of MMP-2, MMP-9, collagen and elastin. All reactions were performed in triplicate, and the relative expression of the specific mRNA was determined with respect to the hypoxanthine guanine phosphoribosyl transferase (HPRT) housekeeping gene. The fold-change of gene expression was calculated by using the comparative threshold method (ΔΔCt = difference of ΔCt between TVE treated cells and control), and the results are expressed as normalized fold expression compared to the controls; this expression was calculated as the ratio of the crossing points of the amplification curves of several genes and an internal standard by using the Bio-Rad iQ™5 software (Bio-Rad Laboratories Srl, Milan, Italy). The protocol was slightly modified from that of Stellavato et al. 2016 [29].

### 4.7. Collagen I, Elastin, Aquaporine 3 and Integrin αV Evaluation by Western Blotting

The HaCaT cells were seeded in 6-well plates, and, at full confluence, the monolayer was scratched and treated with 10%, 5% and 3% TVE for 48 h. The protein lysate from cells were isolated with a Radio Immuno Precipitation assay (RIPA)buffer. Protein concentrations were determined by a protein assay reagent (Bio-Rad. Equal amounts of proteins (20 μg) were loaded by SDS-PAGE on 10% polyacrylamide gel. In the next step, the protein specimens were transferred to a nitrocellulose membrane and probed with a blocking solution for 1 h. The filters were incubated with antibodies against collagen type 1 (goat polyclonal immunoglobulin ( IgG) C-18 sc-8784; 1:250 *v*/*v*), integrin αV (mouse monoclonal IgG P2W7 sc-9969; 1:200 *v*/*v*), AQP3 (mouse monoclonal IgG (F-1): sc-518001; 1:200 *v*/*v*), and actin (actin; goat polyclonal IgG I-19; 1:500) at room temperature for 2 h. Membranes were washed three times for 10 min and incubated with a 1:5000 dilution of horseradish peroxidase-conjugated anti-mouse antibodies and with a 1:10000 dilution of horseradish peroxidase-conjugated anti-goat antibodies for 1 h. All primary antibodies were purchased from Santa Cruz Biotechnology, Dallas, TX, USA, and secondary antibodies were obtained from Bethyl Laboratories (Montgomery, TX, USA). Blots were developed by using the Enhanced Chemiluminescence (ECL) system according to the manufacturer’s protocols (Amersham Biosciences, Little Chalfont Buckinghamshire United Kingdom). An actin antibody was used as the gel loading control. Densitometric analyses were performed using a Gel Doc 2000 UV System and the Gel Doc EZ Imager (Quantity One software, System ID 4020DABD, Bio-Rad Laboratories s.r.l.).

## 5. Conclusions

The present in vitro study, based on human keratinocytes, shows the beneficial effects of TVE (from Farmaceutici Damor, Naples, Italy) at different levels. This study also suggests the molecular basis for TVE efficacy. All doses that were tested on cell proliferation were not cytotoxic and even increased cell viability and proliferation. Regarding scratch repair, TVE increased cell migration in respect to the control, correctly modulating the timing of metalloproteases expression toward a consistent and well-assessed matrix remodeling. Moreover, TVE increased collagen I and elastin expressions, and it positively modulated integrin and acquaporin3; all these biomolecules may concur in a better remodeling of dermal tissue during healing.

## 6. Patents

Riccio, R. (2018). *U.S. Patent No. 9,895,392*. Washington, DC: U.S. Patent and Trademark Office.

## Figures and Tables

**Figure 1 molecules-25-00431-f001:**
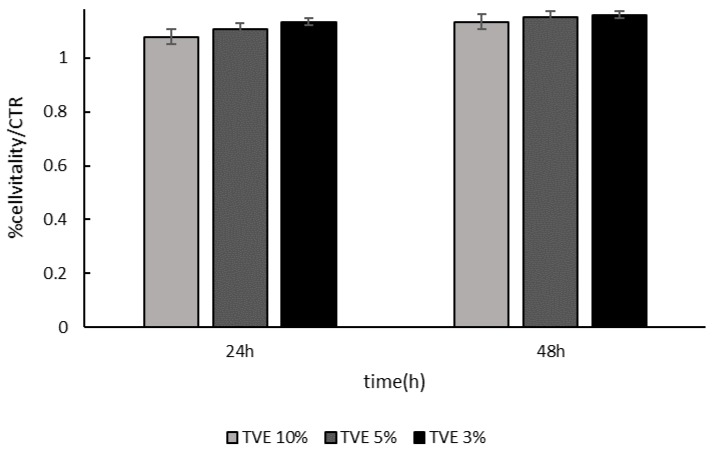
HaCat cell viability measurement. Cells were incubated in the presence of increasing concentrations of *Triticum vulgare* (TVE) (from 3% to 10%) for 24 and 48 h. Cell vitality was normalized in respect to the control.

**Figure 2 molecules-25-00431-f002:**
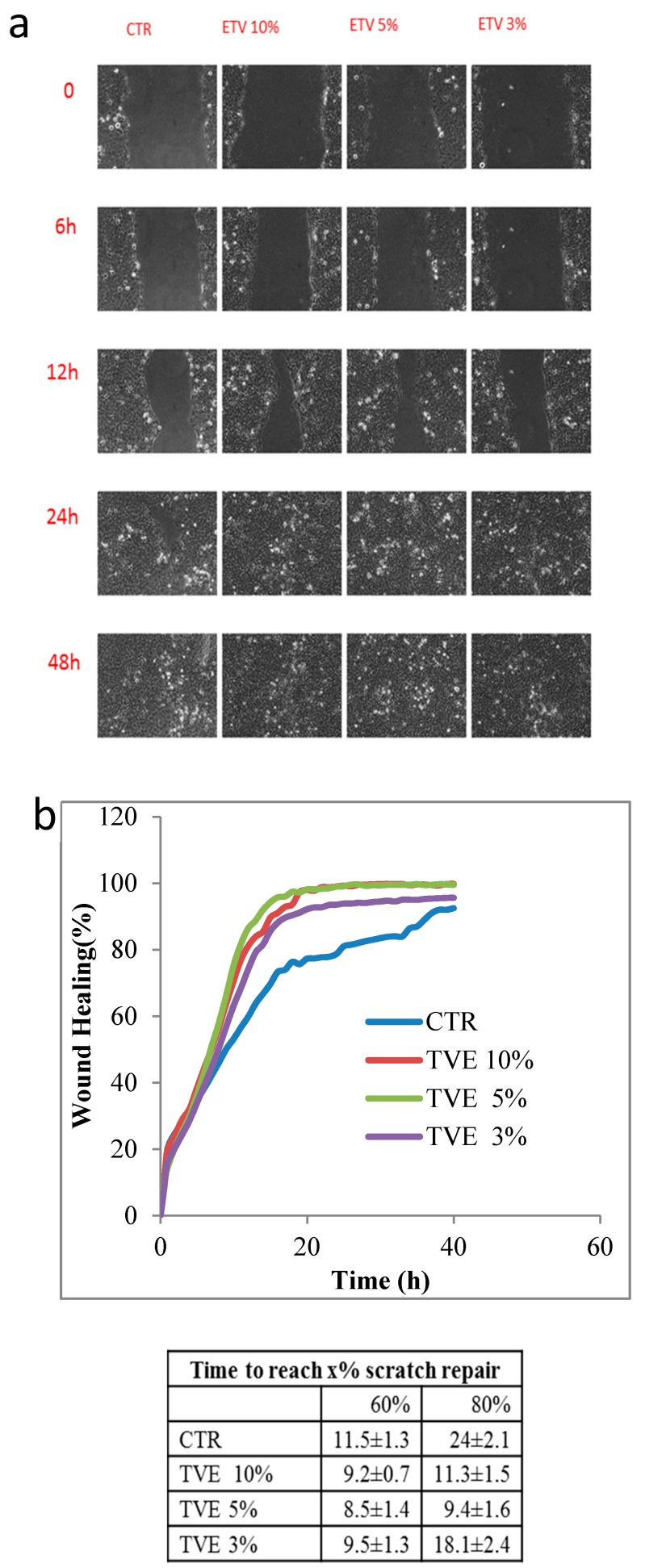
Wound closure analysis: (**a**) Representative panels of images at different times of wound closure in the control and in the presence of treatments (TVE 10%, 5% and 3% *v*/*v*). (**b**) Quantitative analyses of wound closure (Areat0−Area t/Areat0) × 100] vs. time in the CTR and in presence of TVE samples. The analysis was performed on least five fields of view.

**Figure 3 molecules-25-00431-f003:**
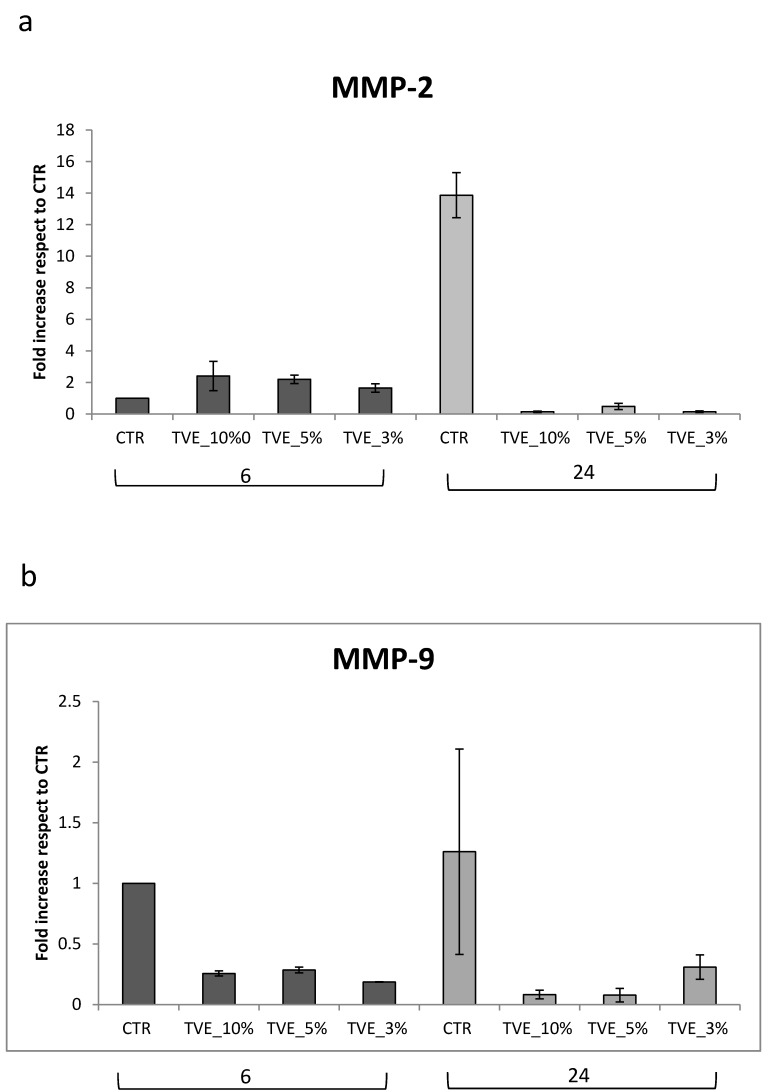
Real-time PCR analysis. At 6 and 24 h, the total RNA was extracted and qRT-PCR was performed to determine the gene expression of MMP-2 (**a**) and MMP-9 (**b**). Data are presented as mean ± SD.

**Figure 4 molecules-25-00431-f004:**
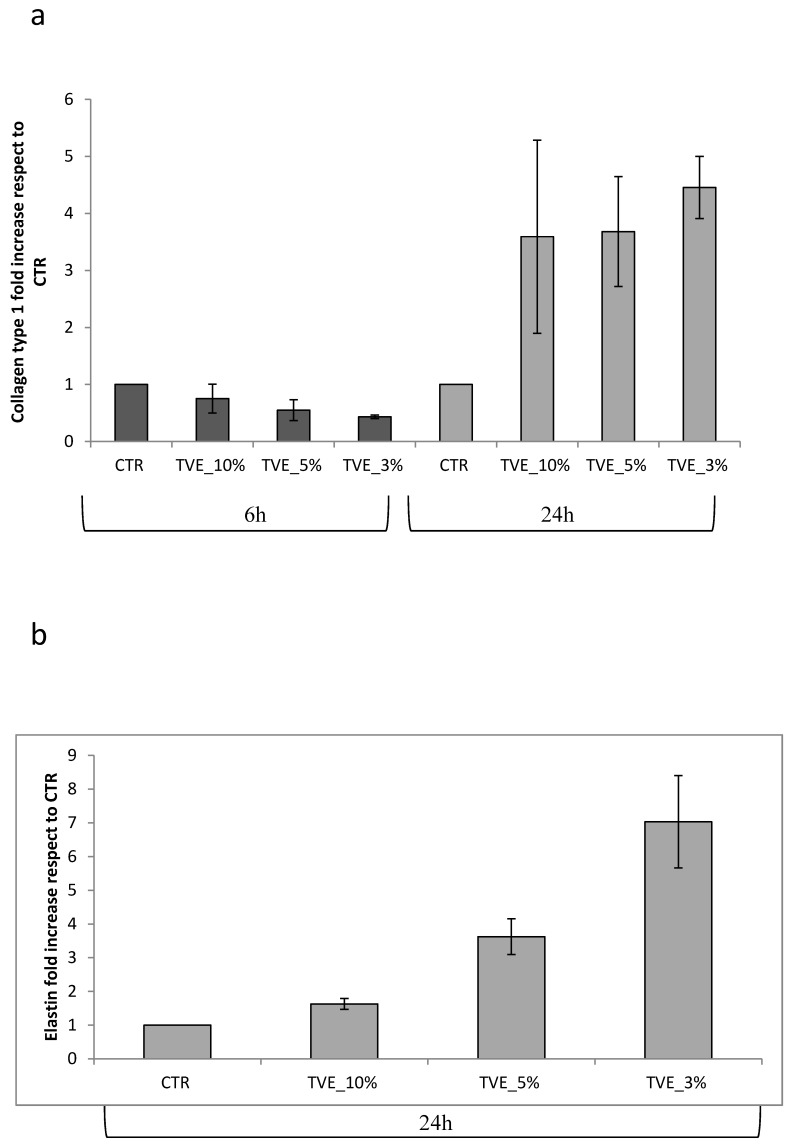
Real-time PCR analysis. The total RNA was extracted and qRT-PCR was performed to determine the gene expression of collagen type I at 6 and 24 h (**a**), as well as elastin expression at 24 h (**b**). Data are presented as mean ± SD.

**Figure 5 molecules-25-00431-f005:**
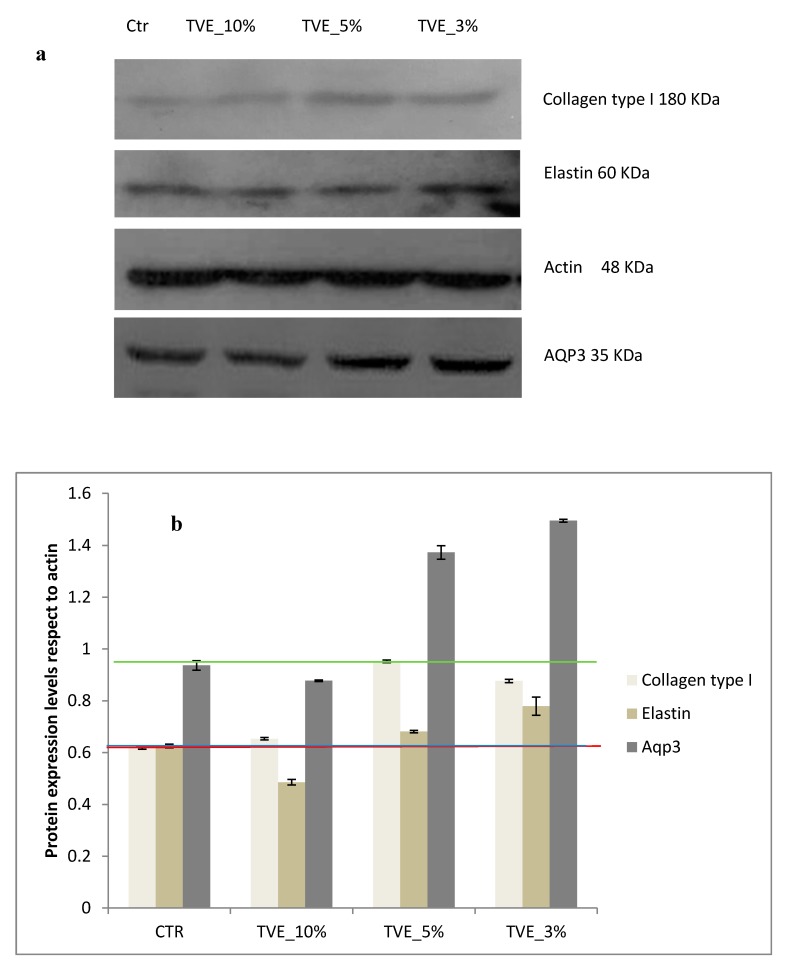
Western blot analysis. (**a**) Protein expression level of collagen type I, elastin and acquaporin 3 AQP3. (**b**) Densitometric results were normalized in respect to actin. All values are expressed in the form of mean ± SD (*n* = 3).

**Figure 6 molecules-25-00431-f006:**
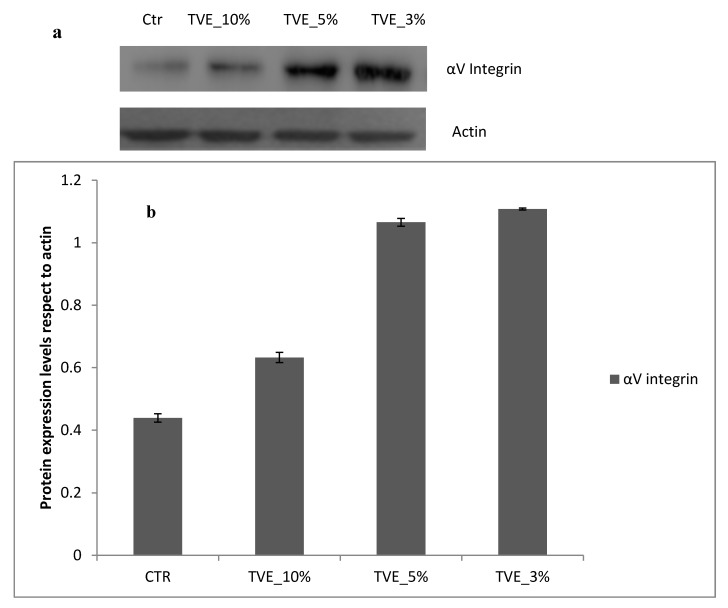
Western blot analysis. (**a**) Protein expression level of integrin αV. (**b**) Densitometric results were normalized in respect to actin. All values are expressed in the form of mean ± SD (*n* = 3).

**Table 1 molecules-25-00431-t001:** Weight average molar mass (M_w_), polydispersity index (M_w_/M_n_), intrinsic viscosity (IV), and representatives are shown. The analyses were performed at least in duplicate.

Peak	M_w_ (kDa)	M_w_/M_n_	IV(dL/g)	Repres (%)
I	6.288 ± 0.432	1.79 ± 0.02	0.030	28.4
II	4.900 ± 0.456	1.78 ± 0.02	0.0278	69.1

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
