# Peer review of "Molecular Mechanisms at the Basis of Pharmaceutical Grade Triticum vulgare Extract Efficacy in Prompting Keratinocytes Healing"

_molecules, 2020, doi:10.3390/molecules25030431_

Round 1

Reviewer 1 Report

The authors Review the efficacy of TVE in a scratch model of keratinocyte migration. The product seems to improve keratinocyte migration in this model. I have some questions:

1) It is hard to believe that no one else has examined keratinocyte migration in a product that is currently on the market. Are you the first?

2) It is hard to understand the relevance of MMP findings in a pure keratinocyte in vitro model. What is the significance of the MMP changes? What do they mean?

3) You also show changes in collagen I levels in the keratinocytes but those cells are not the main producers of collagen in wounds, fibroblasts are. Can you explain the relevance of keratinocyte collagen production in wound healing?

4) There is a great deal of cross-talk between different types of wound healing cells that change the expression of proteins, etc. and affect wound closure. It is hard to find relevance in isolated single cell lines. In addition, you use transformed HaCaT cells. It is hard to know if commercial cell lines are really representative of in vivo human keratinocytes. 

Author Response

The authors Review the efficacy of TVE in a scratch model of keratinocyte migration. The product seems to improve keratinocyte migration in this model. I have some questions:

It is hard to believe that no one else has examined keratinocyte migration in a product that is currently on the market. Are you the first?

To best of our knowledge there are no literature reports on TVE and keratinocytes. We also run a keywords search in the last few days in pubmed and no specific titles or abstracts came up.  (e.g. on 27/12/2019 5 results, no one of extract from Triticum vulgare, only 3 with keratinocytes).

We inserted in the introduction section the following references:

Appl Biochem Biotechnol. 2019 Jun;188(2):381-394. doi: 10.1007/s12010-018-2913-1. Epub 2018 Nov 26.In Vitro Wound Healing Activity of Wheat-Derived Nanovesicles.

Åžahin F1, Koçak P2, GüneÅŸ MY2, Özkan Ä°2, Yıldırım E2, Kala EY2.

Mol Med Rep. 2018 Sep;18(3):3461-3468. doi: 10.3892/mmr.2018.9339. Epub 2018 Jul 31.

Triticum aestivum sprout extract attenuates 2,4‑dinitrochlorobenzene‑induced atopic dermatitis‑like skin lesions in mice and the expression of chemokines in human keratinocytes.

Lee JH1, Ki HH1, Kim DK1, Lee YM2.

However it should also be considered that a new patented process on aqueous extract of TVE has been issued, therefore the product analysed in the framework of this research project may not be exactly the same than the one previously obtained in well consolidated processes.

Line62-63 we insert the following sentence:

Concerning natural products, recent literature reported beneficial effect on dermatological diseases of Triticum aestivum performed on HaCat cellular model [13,14].

It is hard to understand the relevance of MMP findings in a pure keratinocyte in vitro model. What is the significance of the MMP changes? What do they mean?

The reviewer is correct, it is interesting to evaluate co-cultures (we also established an in vitro mode in D’Agostino et al 2019 ). However a simplified assessment of the experimental set up may improve the robustness of results on this specific cell type and make easier to interpreting outcomes in more complex in vitro model, and beyond in vivo. Because recent experimental set up involved human fibroblast we aimed at completing the superficial dermal panel with keratinocytes.

Taking into account the recent literature (Bassino et al. 2019) we inserted in the discussion few sentences on the MMP modulation in HaCaT cells in vitro model.

In fact , the authors found the modulation of MMPs expression is evident for  extract on wound healing process. In fact, that natural compounds containing flavonoids change the MMP-9 expression at all doses tested only during the wound repair and were not modulated in physiological conditions. In our study we investigated also the modulation of MMP-2 at gene level to deepen knowledge on the molecular mechanism involved in wound closure.

Bassino E, Gasparri F, Munaron L.Natural dietary antioxidants containing flavonoids modulate keratinocytes physiology: In vitro tri-culture models.

J Ethnopharmacol. 2019 Jun 28;238:111844. doi: 10.1016/j.jep.2019.111844. Epub 2019 Mar 30.

A sentence was inserted in the discussion section on the specific point

Lines 186-191 page 9:

Recently, Bassino and collaborators, (2019) [23] found plant extracts modulated MMPs expression using scratched HaCaT cells in vitro model. In fact, that natural compounds containing flavonoids changed the MMP-9 expression at all doses tested only during the wound repair and were not modulated in physiological conditions. In our study we investigated also the modulation of MMP-2 at gene level to deepen knowledge on the molecular determinants prompting wound closure towards healthy and functional dermal tissue.

You also show changes in collagen I levels in the keratinocytes but those cells are not the main producers of collagen in wounds, fibroblasts are. Can you explain the relevance of keratinocyte collagen production in wound healing?

The reviewer is correct when appointing that a superior expression of collagen I is obtained by fibroblast. However, in our model we are not comparing the two cell types, we are only evaluating the ability of TVE to positively influence collagen type 1 respect to the control that consisted in only scratched HaCaT. The modulation of this biomarker in HaCaT has been previously reported when the cells were treated with other compounds (e.g Hyaluronic acid, polyphenols, etc.) or devices (Photodynamic therapy). References are reported below, and a sentence was added in the discussion section.

 The expression of collagen type 1 by HaCat cells during wound healing is previously reported by our group in D’Agostino A et al. 2015 (BMC 2015)inserted as reference 29.

The following papers report that keratinocytes produced collagen:

Kwon, K. R., Alam, M. B., Park, J. H., Kim, T. H., & Lee, S. H. (2019). Attenuation of UVB-Induced Photo-Aging by Polyphenolic-Rich Spatholobus Suberectus Stem Extract Via Modulation of MAPK/AP-1/MMPs Signaling in Human Keratinocytes. Nutrients, 11(6), 1341.

Ryu, A. R., & Lee, M. Y. (2017). Chlorin e6-mediated photodynamic therapy promotes collagen production and suppresses MMPs expression via modulating AP-1 signaling in P. acnes-stimulated HaCaT cells. Photodiagnosis and photodynamic therapy, 20, 71-77.

A sentence was inserted in the discussion section on the specific point

Lines 202-205 page 10:

Even though fibroblast are generally appointed as the major collagen producer, few reports evaluated the modulation of this matrix protein in HaCaT cell model. Specifically a superior expression of collagen I  has been previously reported when the cells were treated with other compounds [27,28,29].

There is a great deal of cross-talk between different types of wound healing cells that change the expression of proteins, etc. and affect wound closure. It is hard to find relevance in isolated single cell lines. In addition, you use transformed HaCaT cells. It is hard to know if commercial cell lines are really representative of in vivo human keratinocytes. 

We see the referee point. In fact, we demonstrated in a previous paper that there is a timing in differential expression of proteins by keratinocytes and fibroblast in co-culture (D’Agostino et al 2019). However, this extract was analyzed for the first time on keratinocytes and thus  we tried to simplify the model and be more complete in assessing the biomarkers panel. We found diverse literature reports based on HaCaT in vitro model published in the last 3 years, few of those were inserted as references, as pointed out above.

Reviewer 2 Report

This manuscript descibe the wound healing of natural product extract. This issue may be important. My suggestions are list below.

line 15, Notwithstanding? Please correct this typo. line 80, 81, Table 1. Please use the same form of MW or Mw, also italic or not.

Author Response

This manuscript describe the wound healing of natural product extract. This issue may be important. My suggestions are list below.

line 15, Notwithstanding? Please correct this typo. line 80, 81, Table 1. Please use the same form of MW or Mw, also italic or not.

We corrected the text according to the suggestion of the referee.

Reviewer 3 Report

The paper is well written and the topic clearly described.

However, to facilitate the translation of results into explanation of established clinical practice, I was wondering if the test should be done on primary keratinocytes instead of immortalized cells. It is known that HaCat cells often disply an altered response to growth factors and cytokines. Furthermore, primary keratinocytes are sensitive to calcium concentration whereas HaCaT are relatively insensitive to variations in calcium concentration. Authors should at least comment this point, explaining  the reason to choose immortalized cells.  Explain and comment why in most of results there is not a clear dose-response effects of TVE, in relationship to the considered %.

Author Response

The paper is well written and the topic clearly described.

However, to facilitate the translation of results into explanation of established clinical practice, I was wondering if the test should be done on primary keratinocytes instead of immortalized cells. It is known that HaCat cells often disply an altered response to growth factors and cytokines.

As pointed out above we found few literature reports using HaCaT model even in recent publications.

However we agree with the referee that depending on the phenomenon to be observed HaCat might not be the most consistent model for translational purpouses.

However in this case we do not pursue to analyse cytokines and other molecules that already were shown to be differently modulated in primary keratinocytes and HaCaT

Furthermore, primary keratinocytes are sensitive to calcium concentration whereas HaCaT are relatively insensitive to variations in calcium concentration. Authors should at least comment this point, explaining  the reason to choose immortalized cells. 

The role of calcium was not investigated in our in vitro model. However in literature are reported discordant opinion on its involvement during wound closure. In fact, recent literature (Smits, J. P et al 2017. Scientific reports, 7(1), 11838) reports that an higher sensibility of HaCat of Calcium concentrations respect to primary cells was found with specific reference to some proteins like fillagrin, loricrin and involucrin, however these were not investigated in our model.

Explain and comment why in most of results there is not a clear dose-response effects of TVE, in relationship to the considered %.

We do not notice appreciable difference in wound closure in time lapse experiments, however, at 80% of closure there is significant difference between TVE 3% and the other concentration (5 and 10%).

On the contrary, at gene and protein level we notice difference varying the amount of TVE, for example in Collagen and Elastin gene expression.

At protein level we have an improvement in collagen, elastin and aquaporin expression when the TVE concentrations increased. In addition, integrins amount is higher when the cells were treated with 3 and 5% of TVE. These results confirmed that, even if the wound closure was not extensively influenced by the TVE concentrations,   at molecular level the amount of these novel products have an efficacy in tissue integrity. 

Round 2

Reviewer 1 Report

The authors confirm that the issues I previously brought up are problems. But there have been no substantive changes. The studies are performed well but the relevance of the studies are still a problem.